# Microstructural Properties of Brain White Matter Tracts in Breast Cancer Survivors: A Diffusion Tensor Imaging Study

Tatyana Bukkieva [1,*], Maria Pospelova [1], Aleksandr Efimtsev [1], Olga Fionik [1], Tatyana Alekseeva [1], Konstantin Samochernykh [1], Elena Gorbunova [1], Varvara Krasnikova [1], Albina Makhanova [1], Aleksandra Nikolaeva [1], Samvel Tonyan [1], Anna Lepekhina [1], Anatoliy Levchuk [1], Gennadiy Trufanov [1], Serik Akshulakov [2] and Maxim Shevtsov [1,3,4,*]

1 Personalized Medicine Centre, Almazov National Medical Research Centre, 2 Akkuratova Str., 197341 St. Petersburg, Russia
2 National Center for Neurosurgery, Turan Ave., 34/1, Nur-Sultan 010000, Kazakhstan
3 Department of Radiation Oncology, Technishe Universität München (TUM), Klinikum rechts der Isar, Ismaningerstr. 22, 81675 Munich, Germany
4 Laboratory of Biomedical Nanotechnologies, Institute of Cytology of the Russian Academy of Sciences (RAS), Tikhoretsky Ave., 4, 194064 St. Petersburg, Russia
* Correspondence: tanya-book25@mail.ru (T.B.); shevtsov-max@mail.ru (M.S.); Tel.: +7-999-211-2530 (T.B.); +49-173-148-8882 (M.S.)

**Abstract:** Complex breast cancer (BC) treatment can cause various neurological and psychiatric complications, such as postmastectomy pain syndrome, vestibulocerebellar ataxia, and depression, which can lead to microstructural damage of the white matter tracts of the brain. The purpose of the study is to assess microstructural changes in the white matter tracts of the brain in BC survivors using diffusion tensor imaging (DTI). Single DTI scans were performed on patients ($n = 84$) after complex BC treatment (i.e., surgery, chemotherapy and/or radiation therapy) and on the control group ($n = 40$). According to the results, a decrease in the quantitative anisotropy (FDR $\leq 0.05$) was revealed in the bilateral corticospinal tracts, cerebellar tracts, corpus callosum, fornix, left superior corticostriatal and left corticopontine parietal in patients after BC treatment in comparison to the control group. A decrease in the quantitative anisotropy (FDR $\leq 0.05$) was also revealed in the corpus callosum and right cerebellar tracts in patients after BC treatment with the presence of postmastectomy pain syndrome and vestibulocerebellar ataxia. The use of DTI in patients after BC treatment reveals microstructural properties of the white matter tracts in the brain. The results will allow for the improvement of treatment and rehabilitation approaches in patients receiving treatment for breast cancer.

**Keywords:** breast cancer; breast cancer treatment; post-mastectomy pain syndrome; diffusion tensor imaging; white matter tracts; connectome

## 1. Introduction

Breast cancer (BC) currently remains the most common oncological disease among women worldwide, with 2.26 million new cases of BC in 2020, which accounted for 11.7% of all cancers [1]. In Russia, BC is also in the top position with regard to the incidence of malignant neoplasms of the female population, with 74.87 per 100,000 population (20.4%), the second place in terms of overall cancer incidence (11.1%) [2].

Currently, great progress has been made in the treatment for BC, which traditionally includes surgical treatment followed by chemo- and radiotherapy. In most cases it is carried out in three stages: preoperative (induction) chemotherapy, local (surgery ± radiation therapy) and postoperative treatment (adjuvant therapy).

Surgery remains one of the main and most frequently used methods of BC treatment, and it can be presented in different volumes depending on the stage of the process [3,4].

Recently, there has been a tendency to reduce the use of aggressive surgical techniques, due to the rethinking of clinical and biological concepts of the development of the oncological process, as well as an increase in the number of women with earlier stages of BC [5]. However, one of the most commonly used methods of surgical treatment for BC is still Patey radical mastectomy, which consists of removing the breast, surrounding fat and lymph nodes, as well as removing the small pectoral muscle [6]. Radical mastectomy is a traumatic operation, which, in combination with chemo- and/or radiation therapy, leads to the development of various disorders of the lymphatic, cardiovascular, musculoskeletal and nervous systems in BC survivors [7–10].

Neurological disorders after BC treatment are presented in the form of changes in both peripheral and central nervous systems. Peripheral nervous system damage is usually represented as sensitivity disorders on the inner surface of the shoulder and the axillary fossa and so-called postmastectomy pain syndrome (PMPS) [11–13]. These symptoms are primarily caused by local fibrous-atrophic postoperative and post-radiation changes. However, at a later date, the "centralization" of the chronic pain syndrome occurs with the involvement of structural and functional elements of the "pain connectome" of the brain [14–16]. According to recent data, the prevalence of psychological and psychiatric disorders, such as depressive and anxiety disorders, and insomnia in patients after BC ranges from 29% to 47%, with the development of the major depression disorder in about 10–25% of women after BC treatment [17–20].

A number of studies have reported the effects of adjuvant chemotherapy and radiation therapy on the structure of both white and grey matter of the brain in the form of macro- and microstructural changes in the white matter tracts, as well as the reduction in gray matter density [21–25]. The study by Stouten-Kemperman et al. (2015) compared the effect of different treatment regimens on the structure of the white matter of the brain in patients >10 years after adjuvant chemotherapy and showed white matter damage in a group of patients following high-dose chemotherapy [25]. In another study by Deprez et al., (2012) the chemotherapy-treated group performed significantly worse on attention tests, psychomotor speed and memory tests, which correlated with the significant decrease of fractional anisotropy in frontal, parietal, and occipital white matter tracts [26].

All of the above complications can lead to structural changes in the brain, which worsens the long-term prognosis of rehabilitation and the quality of life of patients. Diffusion tensor imaging (DTI) is a relatively modern MRI technique which assesses microstructural changes of white matter in the brain. This method of imaging allows us to assess the diffusion of water molecules along the white matter tracts of the brain [27–29].

DTI is widely used to detect microstructural damage in white matter tracts in multiple neurological disorders, including chronic pain syndrome, depression, and vestibulocerebellar ataxia [30–33]. The number of studies using the DTI technique in BC patients, including those described above, is attributable to the effect of adjuvant chemotherapy on the structure of the white matter in breast cancer survivors [34–36]. However, there are currently no studies that would evaluate structural changes in the white matter tracts of the brain in patients with various neurological and psychiatric complications of complex BC treatment, such as postmastectomy pain syndrome, vestibulocerebellar ataxia, or depression.

In the current study, the microstructural changes in the white matter tracts of the brain in BC survivors were assessed employing diffusion tensor imaging (DTI). In particular, the quantitative anisotropy was analyzed in patients with the presence of postmastectomy pain syndrome ($n = 71$), vestibulocerebellar ataxia ($n = 43$) and depression ($n = 27$) after BC treatment.

## 2. Materials and Methods

### 2.1. Participants

An open single-center controlled study of microstructural changes in the white matter tracts of the brain in patients after BC treatment was conducted at the Almazov National Medical Research Centre.

A total of 84 female patients after BC treatment (age 30 to 50, mean 45.5 ± 4.1) and 40 healthy female volunteers of the same age category (age 30 to 50, mean 40.5 ± 3.8) were enrolled in the study. All the patients who participated in the study were in remission and had not received treatment for BC at the time of the study.

All patients were in the late postoperative period (>12 months) after Patey radical mastectomy (unilateral or bilateral), eight patients had undergone a combination of surgical treatment and radiation therapy, 27 received a combination of surgical treatment and systemic therapy, and 42 patients received a complex treatment (combination of surgery, radiation therapy, and chemotherapy) (Table 1). The study involved patients who developed various neurologic symptoms related to the treatment itself, but not to the primary cancer disease.

**Table 1.** The number of patients with various methods of BC treatment.

| Treatment Method | Number of Patients |
|---|---|
| Surgery | 7 |
| Surgery + Radiation therapy | 8 |
| Surgery + Systemic chemotherapy | 27 |
| Surgery + Radiation therapy + Systemic chemotherapy | 42 |

### 2.2. Exclusion Criteria

- progression of the main oncological disease;
- the presence of distant metastases (including brain metastases);
- diseases of the brain and cerebral vessels (i.e., brain tumors, demyelination disease, development anomalies, traumatic brain injury, aneurysms and arteriovenous malformations, hemodynamically significant stenoses of the head and neck main arteries and other relevant pathologies);
- decompensated somatic pathology, acute infectious and mental diseases;
- pregnancy;
- contraindications to MRI.

### 2.3. Patients' Examination Data

All patients who participated in the study underwent a standard examination, including the collection of a relevant anamnesis (date of surgery, presence of chemotherapy, radiation therapy) and the assessment of complaints. Particular attention was paid to complaints of edema of the upper limb on the side of surgical treatment, pain in the upper limb and upper arm, sensitivity disorders of the upper limb, paresthesia, muscle weakness, restriction of movement in the shoulder joint, headaches, dizziness, and sleep disorders.

All patients underwent a standard neurological examination with a comparative assessment of the sensitivity and muscle strength of the upper extremities. Additionally, the Adson test was performed for evaluation of thoracic outlet syndrome, as well as hand dynamometry on both sides to assess the strength of the hands and a comparative measurement of the circumference of the hands at five points to assess the presence and degree of edema.

A Visual Analogue Scale (VAS) and McGill questionnaire were used to assess the intensity and the characteristics of chronic pain; a Zung depression scale and STAI anxiety scale were used to determine the presence and degree of anxiety and depressive disorders. Quality of life was assessed using SF-36 and EORTC QLQ-C30 questionnaires.

The study was approved by the ethics committee of Almazov National Medical Research Centre (Protocol number 05112019 from 11.11.2019) and was performed in accordance with the Helsinki Declaration. All subjects provided written informed consent.

## 2.4. Scan Acquisition

MR sequences were obtained employing a SIEMENS TrioTim (3 T) scanner (Siemens, Munich, Germany). Patients underwent MRI of the brain, which included the standard MRI protocol (using T1-, T2-, TIRM, MPRAGE) and DTI. The standard brain MRI protocol was used to exclude the presence of significant CNS pathology in patients after BC treatment and women from the control group.

The diffusion images were acquired using a 2D EPI diffusion sequence, with TR = 4000 ms, and TE = 92 ms. A DTI diffusion scheme was used, and a total of 48 diffusion sampling directions were acquired using the following parameters: b-value—1000 s/mm$^2$, in-plane resolution—1.7 mm, slice thickness—4.5 mm (Table 2).

**Table 2.** Pulse sequence parameters of DTI (2D EPI diffusion sequence).

| | |
|---|---|
| Repetition time/TR | 4000 ms |
| Echo time/TE | 92 ms |
| FoV | 230 mm (128 × 128 matrix) |
| Slice thickness | 4.5 mm |
| Flip angle | 90° |
| Voxel size x (mm), y (mm) | 1.79688 × 1.79688 |
| Study time | 3:42 |

## 2.5. Preprocessing

We used a visual quality inspection in DSI studio to check data quality, including a disposal of distortion and motion artifacts, as well as the elimination of "bad slices". We then set up the mask, the purpose of which was to filter out the background region, increase the reconstruction efficacy, and facilitate further visualization. Further preprocessing of DTI data was automatic.

The accuracy of b-table orientation was examined by comparing fiber orientations with those of a population-averaged template [37]. The diffusion data were reconstructed in the MNI space using q-space diffeomorphic reconstruction to obtain the spin distribution function [38,39]. A diffusion sampling length ratio of 1.25 was used. The output resolution in diffeomorphic reconstruction was 1.79688 mm isotropic. The restricted diffusion was quantified using restricted diffusion imaging [40]. The tensor metrics were calculated using DWI with a b-value of 1000 s/mm$^2$.

## 2.6. Connectometry and Statistical Analyses

Postprocessing of DTI data and comparison between groups was carried out using the DSI Studio software package. The quantitative anisotropy (QA) was extracted as the local connectome fingerprint and used in the connectometry analysis [41].

Quantitative anisotropy is based on generalized q-sampling imaging (GQI). This method relies upon the Fourier transform relation between diffusion distributions and q-space signals. QA is associated with axonal density and it allows to determine axon loss more accurately than fractional anisotropy, as it is less affected to edema and decreases more significantly in axonal loss [42,43].

A filtering-selection approach was employed. The first stage filtered out noisy fibers that have QA values lower than a pre-defined threshold. The purpose of this filtering was to eliminate false fibers and to define the termination locations simultaneously. The second stage selected a fiber orientation for each voxel. If a voxel had multiple fibers, the one that formed the smallest turning angle was selected. The angular threshold was defined according to a prior knowledge of the trajectory curvature. After this filtering-selection process, the propagation direction was calculated.

In the study, a group connectometry analysis (correlational tractography) of the quantitative anisotropy of white matter tracts in patients with various syndromes after BC treat-

ment was performed. The presence/absence of the syndrome (chronic pain/vestibulocerebellar ataxia/depression) was chosen as a variability, and then a correlational tractography analysis in DSI studio was performed to find any tracts correlated with this variable [44]. A nonparametric Spearman correlation was used to derive the correlation. As the default study region for group connectometry we have chosen "whole brain".

A T-score threshold of 2.5 was assigned and tracked using a deterministic fiber tracking algorithm to obtain correlational tractography [45]. The quantitative anisotropy values were normalized. The tracks were filtered by topology-informed pruning with four iterations [46]. An FDR threshold of 0.05 was used to select tracks. To estimate the false discovery rate, a total of 4000 randomized permutations were applied to the group label to obtain the null distribution of the track length. The permutation was applied to subject labels to test results against the permuted condition.

According to the results of correlation tractography, we obtained tracts that had a negative and positive correlation with a given variable. We took into account tracts with a negative correlation, i.e., those in which the quantitative anisotropy was significantly lower in patients after BC treatment in comparison with the control group, or in patients with the syndrome (chronic pain/vestibulocerebellar ataxia/depression) in comparison with patients without the syndrome.

## 3. Results

All patients had various complications after BC treatment, including complications from the peripheral and central nervous system. Chronic pain in the postoperative area ($n = 55$, 65%) and in the upper limb ($n = 60$, 71%) was most often observed in patients, even more often than lymphedema of the upper limb on the side of surgical treatment ($n = 54$, 64%). Other common complications included sensitivity disorders of the upper limb ($n = 52$, 62%), paresthesia ($n = 36$, 43%), muscle weakness ($n = 38$, 45%), restriction of movement in the shoulder joint ($n = 41$, 49%), headaches ($n = 39$, 46%), dizziness ($n = 43$, 51%), and sleep disorders ($n = 30$, 36%).

According to the results of the Visual Analogue Scale (VAS), low-intensity pain was detected in 38 patients (45%), moderate-intensity pain in 41 patients (49%), and severe pain was experienced by five patients (6%). The results of the McGill Pain questionnaire showed a predominance of stabbing, pulling, pressing and aching pain types.

The Adson test for the evaluation of thoracic outlet syndrome was positive in 34 patients (40%). The comparative hand dynamometry revealed a decrease in hand strength on the side of surgical treatment in 38 patients (45%).

According to the STAI anxiety scale, 47 patients (56%) showed high situational anxiety and 51 (61%) showed high personal anxiety. Twenty-seven out of 84 patients (32%) were diagnosed with depression according to the Zung scale, while mild depression was detected in 23 patients (27%), moderate depression in three patients (4%), and severe depression in one patient (1%).

When assessing the quality of life according to the results of the SF-36 questionnaire, the average value of general health in patients was 49.5 (with a minimum value of 21, a maximum of 81), the average value of mental health was 53.2 (with a minimum value of 22, a maximum of 88). According to the EORTC QLQ-C30 quality of life questionnaire, the average value of the overall quality of life indicator was 52.3.

The patients were divided into subgroups depending on the presence of certain clinical syndromes (Table 3).

**Table 3.** The number of patients in groups, depending on the syndromes.

| Syndrome | Number of Patients with the Syndrome | Number of Patients without the Syndrome |
|---|---|---|
| Pain in the upper limb/postoperation area (postmastectomy pain syndrome) | 71 | 13 |
| Vestibulocerebellar ataxia | 43 | 41 |
| Depression | 27 | 57 |

In the current study, an intergroup statistical analysis of the quantitative anisotropy of the white matter tracts between several groups was carried out, including the following:

1. a comparison of quantitative anisotropy of white matter tracts between all patients after BC treatment and the control group;
2. a comparison between patients after BC treatment with the presence of postmastectomy pain syndrome and without it;
3. a comparison between patients with vestibulocerebellar ataxia after BC treatment and without it;
4. a comparison between patients with depression after BC treatment and without depression.

This analysis was carried out in order to assess how a particular syndrome affects the structure of the white matter tracts and how significant these changes are.

### 3.1. DTI Results

### 3.1.1. Patients after BC Treatment in Comparison with the Control Group

According to the results of a comparative analysis of DTI between patients after BC treatment ($n = 84$) and the control group ($n = 40$), there was a negative intergroup correlation, i.e., a decrease in the quantitative anisotropy of the white matter tracts in patients after BC treatment in comparison with the control group (FDR $\leq 0.05$) in the bilateral corticospinal tracts, forceps major, forceps minor and tapetum of corpus callosum, bilateral fornix, right and left cerebellar tracts, left superior corticostriatal tract and left corticopontine parietal tract (Figure 1, Table 4).

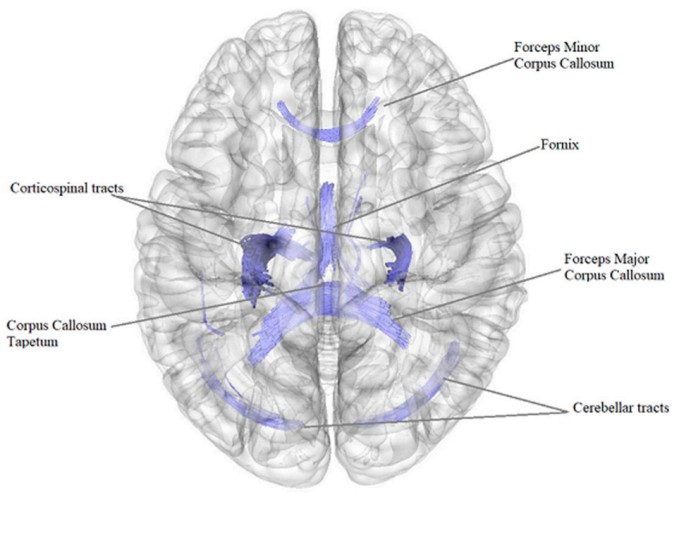

(**a**)

**Figure 1.** *Cont*.

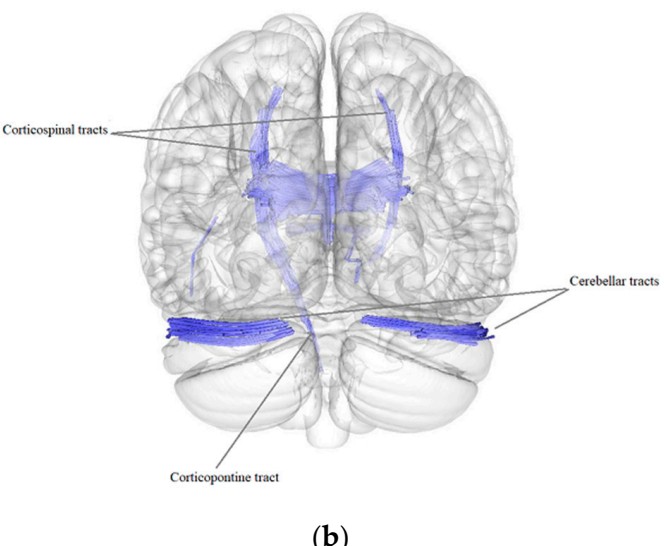

**(b)**

**Figure 1.** Three-dimensional reconstruction of the white matter tracts with the signs of a decrease in quantitative anisotropy in patients after BC treatment in comparison with the control group ((**a**) axial plane, (**b**) coronal plane).

**Table 4.** White matter tracts with lower quantitative anisotropy in patients after BC treatment in comparison with the control group.

| White Matter Tract | Quantitative Anisotropy, Mean Patients after BC Treatment (*n* = 84) | Quantitative Anisotropy, Mean Control Group (*n* = 40) |
|---|---|---|
| Corpus Callosum Tapetum | 0.0696 | 0.0776 |
| Corticospinal tract (Left) | 0.0646 | 0.0734 |
| Fornix (Left) | 0.0502 | 0.0600 |
| Fornix (Right) | 0.0462 | 0.0573 |
| Corpus Callosum Forceps Major | 0.0913 | 0.1005 |
| Corticospinal tract (Right) | 0.0625 | 0.0716 |
| Cerebellum (Right) | 0.0342 | 0.0522 |
| Cerebellum (Left) | 0.0325 | 0.0504 |
| Corpus Callosum Forceps Minor | 0.0681 | 0.0739 |
| Corticostriatal Tract Superior (Left) | 0.0647 | 0.0695 |
| Corticopontine Tract Parietal (Left) | 0.0677 | 0.0722 |

### 3.1.2. Postmastectomy Pain Syndrome

According to the results of a comparative analysis of DTI between patients after BC treatment with the presence of postmastectomy pain syndrome (PMPS) (*n* = 71) and patients after BC treatment without PMPS (*n* = 13), there was a negative intergroup correlation, i.e., a decrease in the quantitative anisotropy of the white matter tracts in patients after BC treatment with the presence of PMPS in comparison with patients after BC treatment

without PMPS (FDR ≤ 0.05) in the forceps major and tapetum of the corpus callosum and in the right cerebellar tracts (Figure 2, Table 5).

### 3.1.3. Vestibulocerebellar Ataxia

According to the results of a comparative analysis of DTI between patients after BC treatment with the presence of vestibulocerebellar ataxia (*n* = 43) and patients after BC treatment without vestibulocerebellar ataxia (*n* = 41), there was a negative intergroup correlation, i.e., a decrease in the quantitative anisotropy of the white matter tracts in patients after BC treatment with the presence of vestibulocerebellar ataxia in comparison with patients after BC treatment without vestibulocerebellar ataxia (FDR ≤ 0.05) in the forceps major and tapetum of corpus callosum and in the right cerebellar tracts (Figure 3, Table 6).

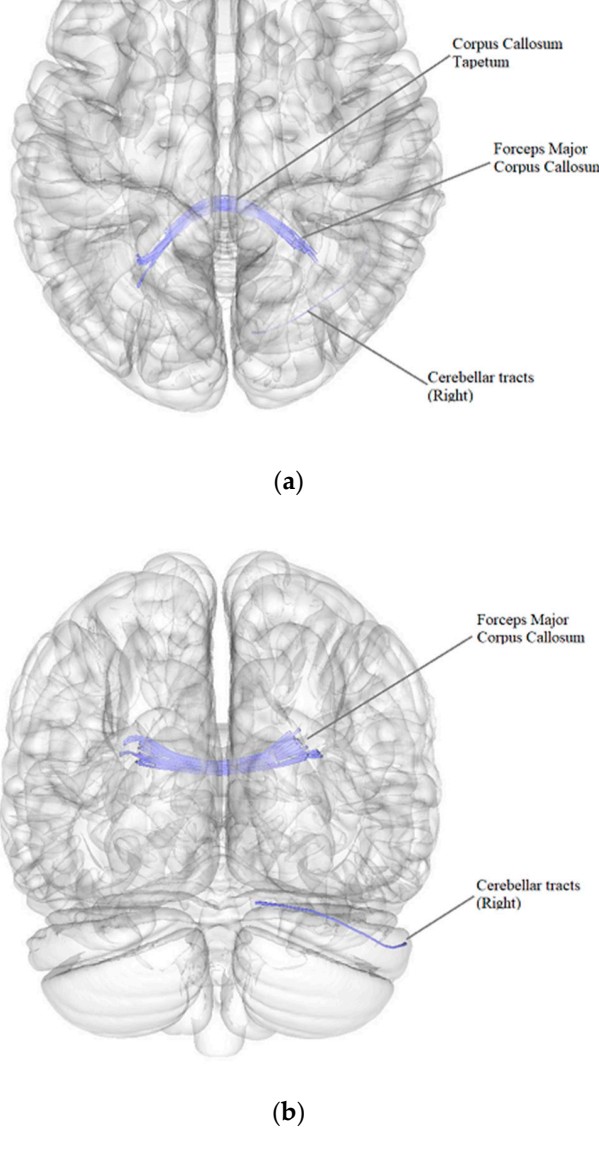

(**a**)

(**b**)

**Figure 2.** Three-dimensional reconstruction of the white matter tracts with the signs of a decrease in quantitative anisotropy in patients after BC treatment with the presence of PMPS in comparison with patients after BC treatment without PMPS ((**a**) axial plane, (**b**) coronal plane).

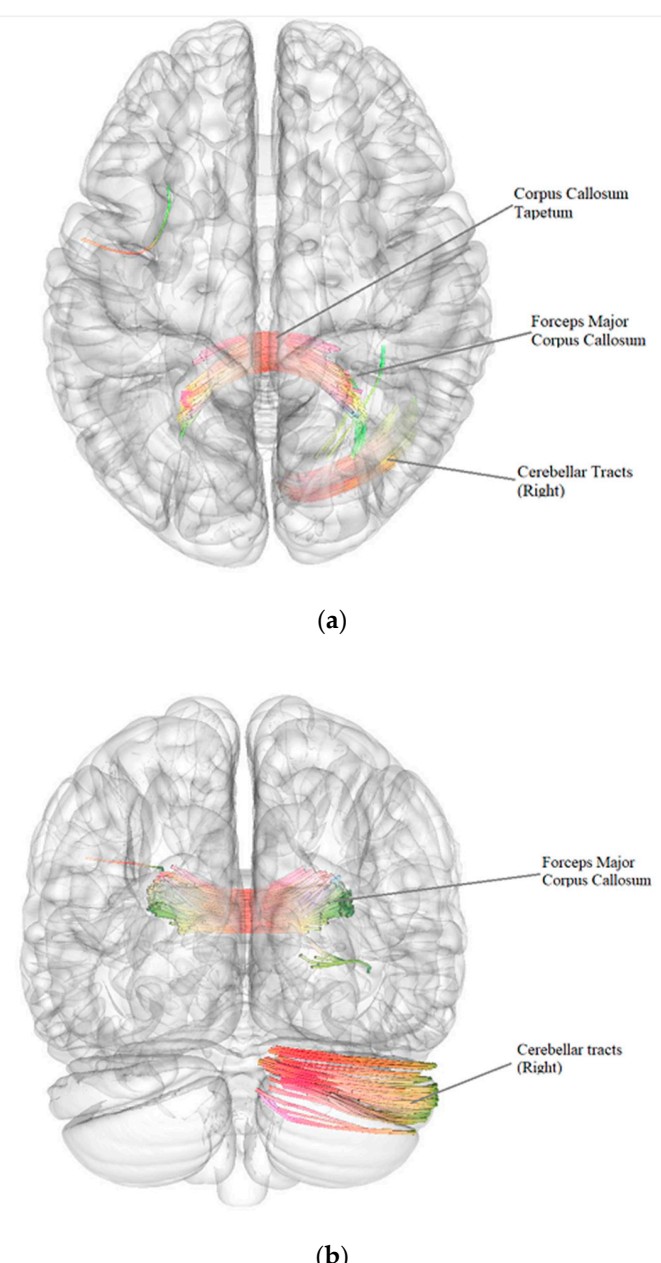

**Figure 3.** Three-dimensional reconstruction of the white matter tracts with the signs of a decrease in quantitative anisotropy in patients after BC treatment with the presence of vestibulocerebellar ataxia in comparison with patients after BC treatment without vestibulocerebellar ataxia ((**a**)—axial plane, (**b**)—coronal plane).

**Table 5.** White matter tracts with lower quantitative anisotropy in patients after BC treatment with the presence of PMPS in comparison with patients after BC treatment without PMPS.

| White Matter Tract | Quantitative Anisotropy, Mean Patients after BC Treatment with PMPS ($n$ = 71) | Quantitative Anisotropy, Mean Patients after BC Treatment without PMPS ($n$ = 13) |
| --- | --- | --- |
| Corpus Callosum Forceps Major | 0.0504 | 0.0563 |
| Corpus Callosum Tapetum | 0.0560 | 0.0588 |
| Cerebellum (Right) | 0.0625 | 0.0687 |

**Table 6.** White matter tracts with lower quantitative anisotropy in patients after BC treatment with the presence of vestibulocerebellar ataxia in comparison with patients after BC treatment without vestibulocerebellar ataxia.

| White Matter Tract | Quantitative Anisotropy, Mean Patients after BC Treatment with Vestibulocerebellar Ataxia (*n* = 43) | Quantitative Anisotropy, Mean Patients after BC Treatment without Vestibulocerebellar Ataxia (*n* = 41) |
|---|---|---|
| Corpus Callosum Forceps Major | 0.0523 | 0.0595 |
| Corpus Callosum Tapetum | 0.0546 | 0.0628 |
| Cerebellum (Right) | 0.0452 | 0.0564 |

### 3.1.4. Depression

When performing an intergroup connectometric analysis of DTI, there were no significant differences in quantitative anisotropy in patients after BC treatment with depression in comparison with patients after BC treatment without depression.

## 4. Discussion

Our results demonstrated the possibilities of using the DTI technique in analyzing the relationship between white matter tracts and various neurological conditions associated with BC treatment. According to the literature, in patients in the long-term period after complex BC treatment (surgical treatment, chemotherapy, radiation therapy), significant changes in the structure of the white matter of the brain are noted both in the form of a general decrease in fractional anisotropy and in the form of a decrease in fractional anisotropy in individual white matter tracts of the brain [25,26,34–36,47,48]. Some studies have identified specific white matter tracts in which there is a decrease in fractional anisotropy in patients after BC treatment. Thus, in a study by Mo et al., (2017) there was a decrease in fractional anisotropy in the fornix and superior fronto-occipital fasciculus in patients after chemotherapy for BC [47]. Nevertheless, there are currently a small number of studies evaluating the quantitative anisotropy of white matter tracts in BC survivors and ones that take into account various neurological complications of BC treatment, such as postmastectomy pain syndrome, vestibulocerebellar ataxia and depression.

According to our data, in patients after BC treatment in comparison to the control group, there was a decrease in the quantitative anisotropy in the various white matter tracts, including tracts of the corpus callosum, fornix, bilateral corticospinal and cerebellar tracts, left corticopontine, and left corticostriatal tracts. Our study showed a decrease in the quantitative anisotropy in the corpus callosum in all patients after BC treatment compared to the control group, as well as in patients with postmastectomy pain syndrome and in patients with vestibulocerebellar ataxia. Chemotherapy-induced microstructural changes in the white matter tracts of the corpus callosum have been shown previously in a large number of studies in patients after chemotherapy for BC [23,24,26,48]. The mechanisms of involvement in the pathological process white matter tracts of the corpus callosum are unknown. However, there is an extensive group of disorders related to the so-called cytotoxic lesions of the corpus callosum (CLOCCs), which can be induced, among other things, by chemotherapeutic agents. In the examined patients, the presence of macrostructural signs of CLOCCs (hyperintense signal in the corpus callosum on T2-WI and TIRM, diffusion restriction on DWI) was excluded by the standard MRI protocol. However, the absence of macrostructural focal changes does not exclude microstructural damage to the white matter tracts of the corpus callosum caused by chemotherapy. The preferred involvement of the corpus callosum in the pathogenesis of cytotoxic brain damage can be explained by the presence of a high density of oligodendrocytes expressing a large number

of glutamate receptors, which contributes to a greater susceptibility of glial cells of this localization to the development of glutamate-induced excitotoxicity [49].

When analyzing the data of DTI in patients after BC treatment compared to the control group, a decrease in quantitative anisotropy in the fornix was revealed. The fornix is the main efferent tract of the hippocampus, connecting it to the mamillary bodies and the anterior nuclei of the thalamus. Despite the fact that the specific function of the fornix is not fully defined at the moment, it is believed that it plays a key role in some aspects of long-term memory, in particular in the regulation of episodic memories [50]. It has been shown that a decrease in the volume of the fornix is an early predictor of cognitive deficits in healthy elderly individuals [51]. Changes in the quantitative anisotropy of the tracts in the fornix have been identified in a number of studies in patients with Alzheimer's disease [52,53]. A decrease in fractional anisotropy in the fornix was detected in patients with BC after chemotherapy, which correlated with a decrease in cognitive tests for long-term memory [54]. Thus, microstructural changes in the white matter of the fornix may be a predictor of early cognitive impairment in patients after chemotherapy treatment for BC.

In our study, a decrease in the quantitative anisotropy in bilateral corticospinal tracts in all patients after BC treatment compared to the control group was revealed. Corticospinal tracts are descending white matter tracts extending from the motor cortex down to the synapse with motor neurons of the spinal cord in the anterior horns, participating in the voluntary movement of the muscles of the body. The bilateral decrease in the quantitative anisotropy of the corticospinal tracts correlates with previous studies of changes in the white matter tracts of the brain in patients after complex BC treatment, and may be primarily caused by the toxic effect of chemotherapy [24,34,36].

In patients after BC treatment in comparison with the control group, a decrease in quantitative anisotropy was also revealed in the parietal part of the left corticopontine tract. Corticopontine tracts are a collective term for white matter tracts that originate in the main parts of the cerebral cortex and spread caudally through the internal capsule to the nuclei of the pons and then continue into the pontocerebellar tracts, connecting various areas of the cerebral hemispheres with the opposite hemisphere of the cerebellum through the middle cerebellar crus. Thus, these tracts ensure the coordination of complex motor functions. Microstructural changes in the corticospinal and corticopontine tracts may indicate a decrease in motor control of the cortex over the underlying parts of the brain, including the spinal cord and cerebellum. Involvement of the parietal part of the corticopontine tracts may also be due to the toxic effects of chemotherapy, and, according to previous studies, may correlate with the initial cognitive deficit in patients [54].

Microstructural changes of the white matter tracts of the cerebellum were observed in all patients after BC treatment (bilaterally), as well as in groups of patients with postmastectomy pain syndrome and vestibulocerebellar ataxia (on the right side). Microstructural changes in the cerebellar tracts may be associated with both the effect of chemotherapy and the presence of chronic ischemia in the vertebral-basilar basin in patients with postmastectomy pain syndrome and vestibulocerebellar ataxia, as a result of the postoperative fibrous-atrophic changes in the brachial neurovascular bundle, and require further study.

It should be noted that in our study, when analyzing the DTI data, no significant changes in white matter tracts were revealed in patients after BC treatment with depression in comparison with patients after BC treatment without depression, which requires further observation and analysis with an expansion of the number of patients.

## 5. Limitations

This study has some limitations. The first limitation is the heterogeneity of the main group in relation to the various methods of BC treatment received by the patients. Furthermore, in the three groups of patients divided by complications (postmastectomy pain syndrome, vestibulocerebellar ataxia, and depression) there were patients that had more than one complication, and that should be taken into account. Therefore, we aim to conduct

a more reliable study on a larger sample of patients in the future in order to overcome these limitations.

In our research, we used deterministic tractography for group connectometry analysis in the DSI studio. However, a number of recent works discuss the advantages of using probabilistic tractography in structural connectome studies [55–57]. This issue needs further research and a comparative analysis of deterministic and probabilistic approaches in tractography.

## 6. Conclusions

The use of DTI expands the understanding of the pathogenesis of the possible neurological effects of BC treatment. Our study revealed a decrease in the quantitative anisotropy in various white matter tracts of the brain in patients after BC treatment and in patients after BC treatment with the manifestation of postmastectomy pain syndrome and vestibulocerebellar ataxia.

In the future it is planned to expand the number of patients and study the effects of certain treatment methods, such as adjuvant chemotherapy, radiation therapy, and different kinds of surgery on the structure of the white matter tracts of the brain. A further clinical and neuroimaging comparison should also be carried out between the neurological and psychiatric complications of BC treatment and the presence of microstructural damage in the white matter tracts of the brain in breast cancer survivors. DTI data may be an important marker for assessing the risk of cognitive impairment after BC treatment [55]. Further research should shed light on whether a decrease in the quantitative anisotropy in certain white matter tracts can be used as a predictor of early cognitive decline in patients after BC treatment.

**Author Contributions:** Conceptualization, T.B., M.P., A.E. and M.S.; methodology, T.B., M.P., A.E. and M.S.; validation, T.B., M.P., A.E., O.F., T.A., K.S. and G.T.; formal analysis, T.B., M.P., A.E., A.L. (Anna Lepekhina), O.F., T.A., K.S., V.K., E.G., A.L. (Anatoliy Levchuk), A.M. and G.T.; investigation, T.B., M.P., A.E., E.G., A.M., A.N., S.T., A.L. (Anatoliy Levchuk), S.A. and M.S.; resources, M.P., A.E., O.F., T.A., K.S., S.A. and M.S.; data curation, M.P., A.E. and G.T.; writing—original draft preparation, T.B., A.L. (Anna Lepekhina) and M.S.; writing—review and editing, T.B., M.P., A.E., S.A. and M.S.; visualization, T.B., E.G., A.N., S.T. and A.L. (Anatoliy Levchuk); supervision, M.P., A.E. K.S., G.T. and M.S.; project administration, M.P., A.E. and M.S.; funding acquisition, M.P., K.S., G.T. and M.S. All authors have read and agreed to the published version of the manuscript.

**Funding:** This work was financially supported by the Ministry of Science and Higher Education of the Russian Federation (Agreement No. 075-15-2022-301 from 20 April 2022). S.A. was supported by the program "Development of a program of molecular-cytogenetic studies and the creation of a biobank of tumors of the central nervous system" (BR10965225) (Ministry of Education and Science of the Republic of Kazakhstan).

**Institutional Review Board Statement:** The study was carried out in compliance with the principles of the Helsinki Declaration of the World Medical Association with the consent of the Ethics Committee of the Federal State Budgetary Institution "Almazov National Medical Research Center" of the Ministry of Health of the Russian Federation (conclusion of 31 October 2019).

**Informed Consent Statement:** Informed consent was obtained from all subjects involved in the study.

**Data Availability Statement:** Not applicable.

**Conflicts of Interest:** The authors declare that they have no conflict of interest.

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
