# Peer review of "Microstructural Properties of Brain White Matter Tracts in Breast Cancer Survivors: A Diffusion Tensor Imaging Study"

_pathophysiology, doi:10.3390/pathophysiology29040046_

Round 1

Reviewer 1 Report

The authors present an interesting study when they look at the microstructural properties of white matter in breast cancer survivors using diffusion-derived metrics. The results are shown in an elegant way, and the introduction and the discussion are nicely framed. However, the methods are confusing.

Methods:

how many b0 (b=0s/mm2) were acquired?

The acquisition is highly anisotropic, this should be mentioned in the limitations.

The paragraph starting in line 150, should be in the following section, in Connectometry and Statistical analyses. 

How the accuracy of the b-table was tested? visually comparing against tracts in template space? Could the authors provide more details on this?

Why are values lower than 1750 used for DTI metrics? is not 1000 the maximum b value?

Quantitative anisotropy is a new measure, could you provide more details about its calculation? maybe it can be useful to compare it with the more standard fractional anisotropy.

Is not clear to me how many measures are the authors computing: DTI measures, restricted water measures, and quantitative anisotropy? If so, why is only reported quantitative anisotropy? The methods need a little bit of clarification in general.

To create the tracts, all the filtering project is explained, but how the tracts are created? It was whole brain tractography and from there the tracts of interest were selected?

Results:

Line 224: 0.050000 should be changed to 0.05 (same every time that appears in the manuscript)

Could the authors explain the color code of figures 1,2 and 3? First I thought that they were showing the p-value in the tracts, but then figure 3 seems to be colored by direction.

Discussion: 

Here quantitative anisotropy is referred to as quantitative fractional anisotropy. Change for consistency.

Conclusion:

The authors talk about DTI here, the meaning of this is that quantitative anisotropy is fractional anisotropy. Is this right?

Author Response

Dear Reviewer,

On behalf of my colleagues, I want to express my gratitude for the time you have devoted to reviewing our article, “Microstructural Properties of Brain White Matter Tracts in Breast Cancer Survivors: A Diffusion Tensor Imaging Study” (pathophysiology-1922725).

We have studied the review carefully and tried our best to revise and improve the manuscript.

Here are the answers to your comments.

  1. how many b0 (b=0s/mm2) were acquired?

Answer: The standard b-value for all DTI scan was 1000 s/mm²

  1. How the accuracy of the b-table was tested? visually comparing against tracts in template space? Could the authors provide more details on this?

Answer: The accuracy of b-table orientation was examined by comparing fiber orientations with those of a population-averaged template (line 385-386). This was done automatically as a part of processing DTI data in the DSI studio.

  1. Quantitative anisotropy is a new measure, could you provide more details about its calculation? maybe it can be useful to compare it with the more standard fractional anisotropy. Is not clear to me how many measures are the authors computing: DTI measures, restricted water measures, and quantitative anisotropy? If so, why is only reported quantitative anisotropy? The methods need a little bit of clarification in general.

Answer: (line 389-393): Quantitative anisotropy is based on generalized q-sampling imaging (GQI). This method relies upon the Fourier transform relation between diffusion distributions and q-space signals. QA is associated with axonal density and it allows to determine axon loss more accurately than fractional anisotropy, as it is less affected to edema and decreases more significantly in axonal loss [42, 43].

  1. To create the tracts, all the filtering project is explained, but how the tracts are created? It was whole brain tractography and from there the tracts of interest were selected?

Answer: In our study, we performed a group connectometry analysis of the quantitative anisotropy of white matter tracts in patients with various syndromes after BC treatment. We have chosen the presence / absence of the syndrome (chronic pain / vestibulocerebellar ataxia / depression) as a variability in analysis and then performed intergroup analysis in DSI studio to find any tracks correlated with this variable.

As the default study region for group connectometry we have chosen “whole brain”, which means that connectometry “looked” at the whole-brain region. “Whole-brain” region was the most suitable in our case, since we wanted to get a general idea of those white matter tracts in which there are differences in quantitative anisotropy between patients with and without the corresponding syndrome, or between patients after BC treatment and the control group.

According to the results of correlation tractography, we obtained tracts that had a negative and positive correlation with a given variable. We took into account tracts with a negative correlation, i.e. those in which the quantitative anisotropy was significantly lower in patients with the syndrome than in patients without the syndrome (data on quantitative anisotropy are shown in the tables).

I have added some of this explanation to the section “Connectometry and Statistical analyses” in the text of an article.

  1. Could the authors explain the color code of figures 1,2 and 3? First I thought that they were showing the p-value in the tracts, but then figure 3 seems to be colored by direction.

Answer: The images in the article show white matter tracts that had lower quantitative anisotropy in patients with the presence of the syndrome in comparison with patients without the syndrome. In this case, the color itself did not play a significant role and did not mean the direction of the tract. However, the intensity of the color and the thickness of the tracts correlate with the difference in quantitative anisotropy (the greater the difference between the groups, the brighter and wider the tract)

6. Discussion: 

Here quantitative anisotropy is referred to as quantitative fractional anisotropy. Change for consistency.

7. Conclusion:

The authors talk about DTI here, the meaning of this is that quantitative anisotropy is fractional anisotropy. Is this right?

Answer: These were corrected.

Reviewer 2 Report

The manuscript focused on an intersting topic and was generally written well. Below I have some concerns and suggestions to improve the manuscript.

The authors mentioned that "Control group enrolled 40 healthy female volunteers of the same age category". However, it is necessary to give more details about the average age in the patients and controls groups. If there is a significant difference in average age between the two groups, it would be better to control age effects in all analyses.

In the Methods section, more details about the DTI scanning parameters should be provided, e.g., the field of view (FOV). Author, more details about how to preprocess the imaging data (e.g., head motion correction) and how to control the data quality should be provided.

In the present study, the quantitative anisotropy of a number of white matter tracts were compared between groups. However, how did the authors define these white matter tracts? Maybe it was based on an Atlas but I did't see any descriptions.

The authors used deterministic tractography in this study; however, probabilistic tractography has been suggested to be more accurate than deterministic tractography in recent years. This might be considered as a limitation and some recent studies using deterministic tractography should be cited, such as: https://www.frontiersin.org/articles/10.3389/fpsyt.2018.00391/full.

Author Response

Dear Reviewer,

On behalf of my colleagues, I want to express my gratitude for the time you have devoted to reviewing our article, “Microstructural Properties of Brain White Matter Tracts in Breast Cancer Survivors: A Diffusion Tensor Imaging Study” (pathophysiology-1922725).

We have studied the review carefully and tried our best to revise and improve the manuscript.

Here are the answers to your comments.

  1. The authors mentioned that "Control group enrolled 40 healthy female volunteers of the same age category". However, it is necessary to give more details about the average age in the patients and controls groups. If there is a significant difference in average age between the two groups, it would be better to control age effects in all analyses.

Answer: Thank you for the important comment. I added to the article the average age of patients after BC treatment and women from the control group (line 114-115). The age of women from the control group was younger on average, but the difference is insignificant and may not be taken into account (45,5±4,1 vs 40,5±3,8).

  1. In the Methods section, more details about the DTI scanning parameters should be provided, e.g., the field of view (FOV). Author, more details about how to preprocess the imaging data (e.g., head motion correction) and how to control the data quality should be provided.

Answer: I have added a table with more detailed DTI parameters (line 379) and a paragraph in which I described in more detail the procedure for preprocessing DTI data using DSI studio software.

  1. In the present study, the quantitative anisotropy of a number of white matter tracts were compared between groups. However, how did the authors define these white matter tracts? Maybe it was based on an Atlas but I did't see any descriptions.

Answer: In our study, we performed a group connectometry analysis of the quantitative anisotropy of white matter tracts in patients with various syndromes after BC treatment. We have chosen the presence / absence of the syndrome (chronic pain / vestibulocerebellar ataxia / depression) as a variability in analysis and then performed intergroup analysis in DSI studio to find any tracts correlated with this variable.

As the default study region for group connectometry we have chosen “whole brain”, which means that connectometry “looked” at the whole-brain region. “Whole-brain” region was the most suitable in our case, since we wanted to get a general idea of those white matter tracts in which there are differences in quantitative anisotropy between patients with and without the corresponding syndrome, or between patients after BC treatment and the control group.

According to the results of correlation tractography, we obtained tracts that had a negative and positive correlation with a given variable. We took into account tracts with a negative correlation, i.e. those in which the quantitative anisotropy was significantly lower in patients after BC treatment in comparison with control group, or in patients with the syndrome (chronic pain / vestibulocerebellar ataxia / depression) in comparison with patients without the syndrome.

I have added some of this explanation to the section “Connectometry and Statistical analyses” in the text of an article.  

  1. The authors used deterministic tractography in this study; however, probabilistic tractography has been suggested to be more accurate than deterministic tractography in recent years. This might be considered as a limitation and some recent studies using deterministic tractography should be cited, such as: https://www.frontiersin.org/articles/10.3389/fpsyt.2018.00391/full.

Answer: Thank you very much for the interesting remark. When using the method of correlation tractography, the DSI Studio provides the possibility to choose deterministic tractography for group connectometry analysis. I have not found any data on the possibility of using probabilistic tractography in this case, however, a number of articles actually show the advantage of this method in processing DTI data. I have indicated this in the Limitations section; however, I believe that this is a debatable issue that requires further discussion.